# Quantifying the Strain: A Global Burden of Disease (GBD) Perspective on Musculoskeletal Disorders in the United States Over Three Decades: 1990–2019

**DOI:** 10.3390/jcm13226732

**Published:** 2024-11-08

**Authors:** Yazan A. Al-Ajlouni, Omar Al Ta’ani, Sophia Zweig, Ahmed Gabr, Yara El-Qawasmi, Godstime Nwatu Ugwu, Zaid Al Ta’ani, Mohammad Islam

**Affiliations:** 1School of Medicine, New York Medical College, Valhalla, NY 10595, USA; 2Department of Physical Medicine and Rehabilitation, Metropolitan Hospital, New York, NY 10029, USA; 3Alleghany Health Network, Pittsburgh, PA 15212, USA; 4SUNY Downstate College of Medicine, Brooklyn, NY 11225, USA; 5Faculty of Dentistry, University of Jordan, Amman 11942, Jordan; 6Faculty of Dentistry, University of Toronto, Toronto, ON M5S3H2, Canada; 7Department of Special Surgery and Orthopedics, University of Jordan Hospital, Amman 11942, Jordan

**Keywords:** musculoskeletal disorders, DALYs (disability-adjusted life years), epidemiology, risk factors, United States

## Abstract

**Background:** Musculoskeletal (MSK) disorders significantly contribute to global disability, especially in high-income countries. Yet, comprehensive studies on their epidemiological burden in the United States (US) are limited. Our study aims to fill this gap by characterizing the MSK disease burden in the US using Global Burden of Disease (GBD) data from 1990 to 2019. **Methods:** We conducted an ecological study using descriptive statistical analyses to examine age-standardized prevalence and disability-adjusted life years (DALY) rates of MSK disorders across different demographics and states. The study also assessed the impact of risk factors segmented by age and sex. **Results:** From 1990 to 2019, the burden of MSK disorders in the US increased significantly. Low back pain was the most prevalent condition. Age-standardized prevalence and DALY rates increased by 6.7% and 17.6%, respectively. Gout and other MSK disorders saw the most significant rise in DALY rates. Females experienced higher rates than males, and there were notable geographic disparities, with the District of Columbia having the lowest and North Dakota and Iowa the highest DALY rates. Smoking, high BMI, and occupational risks emerged as primary risk factors. **Conclusions:** Our study highlights the escalating burden of MSK disorders in the US, revealing significant geographic and sex disparities. These findings highlight the urgent need for targeted health interventions, policy formulation, and public health initiatives focusing on lifestyle and workplace modifications. Region- and sex-specific strategies are crucial in effectively managing MSK conditions, considering the influence of various risk factors.

## 1. Introduction

Musculoskeletal (MSK) disorders are a significant cause of physical disability and a main driver of noncommunicable disease-related disability burden [1]. Globally, MSK disorders make up roughly one-fifth (21.3%) of the total lived years with disability (YLD) and 6.7% of the total disability-adjusted life years (DALY), according to the Global Burden of Disease (GBD) 2010 Study [2,3]. MSK disorders represent the fourth greatest burden on global population health, and the third in high-income countries [4]. In 2019, neck pain was the most common MSK disorder worldwide, affecting 15% of the population, while osteoarthritis was the second most common, affecting 4%. The most prevalent MSK conditions globally in 2017 were low back pain (36.8%), followed by other musculoskeletal disorders (21.5%), osteoarthritis (OA) (19.3%), neck pain (18.4%), gout (2.6%), and rheumatoid arthritis (RA) (1.3%); these proportions have not significantly changed since 1990 [5]. MSK disorders account for a significant proportion of years of life lost (YLLs) from premature death, especially RA, OA, and low back pain [6]. The burden of MSK disease varies by country income levels, with a higher proportion of prevalent cases in high-income countries [6].

According to the GBD study, MSK conditions were among the leading causes of DALYs in high-income countries in 2019. Among all causes of DALYs in females, low back pain was the highest leading cause, other MSK conditions were the 5th, OA was the 14th, and neck pain was the 19th highest. Among all causes of DALYs in males, low back pain was the 3rd highest leading cause, other MSK conditions were the 12th, OA was the 24th, and neck pain was the 25th highest [7].

The burden of MSK conditions is particularly high in the United States (US). The most recent data available from the CDC finds that in 2019, 39.0% of American adults had back pain, 36.5% had lower limb pain, and 30.7% had upper limb pain in the last 3 months. The prevalence of back pain was slightly higher in females (40.6%) than in males (37.2%). The prevalence of back pain increased by age group: 28.4% in adults aged 18–29, 35.2% in ages 30–44, 44.3% in ages 45–64, and 45.6% in ages 65 and over [8]. From 2019 to 2021, 21.2% of adults in the US had ever been diagnosed with arthritis, rheumatoid arthritis, gout, lupus, or fibromyalgia. The prevalence of arthritis was higher in females (24.2%) than in males (17.9) and increased with age (5% in adults aged 18–44, 26% in ages 45–64, and 47% in ages 65 or older) [9].

The economic burden of MSK conditions in the US is the highest of all medical conditions. Specifically, low back and neck pain represented the highest amount of healthcare spending (estimated at USD 134.5 billion) in 2016. Other musculoskeletal disorders accounted for the second highest amount of healthcare spending at USD 129.8 billion [10]. In 2019, the all-cause medical cost of back pain was USD 365 billion, and OA was USD 460 billion [11]. The cost of treating MSK conditions includes federal disability support, which in 2015 totaled USD 868 billion in 2015, making up 36% of all healthcare expenditures for adults [12].

MSK conditions severely restrict people’s activities and participation in society, limiting the ability to work, retire early, and increasing the need for social and financial support [6]. The burden of disability from MSK conditions is predicted to increase due to increasing population, aging, and work- and behavior-related risk factors such as sedentary lifestyle, obesity, and injury [6]. Despite this large burden of disease and financial cost, few studies have characterized the epidemiological burden of MSK conditions in the United States, with research mostly limited to the burden of specific conditions [11,13,14,15,16,17]. The low number of studies on MSK disorders overall may be due to MSK disorders being perceived as rarely fatal and mostly irreversible [6]. Characterizing the burden of MSK disorders in the USA is important in order to better understand a major burden of disease in this country and any trends over time.

The GBD study is a crucial tool for understanding disease burden on a global scale. This dataset provides a comprehensive assessment of disease burden through estimates of multiple measures such as deaths, incidence, prevalence, YLLs, YLDs, and DALYs due to a total of 369 diseases and injuries across 87 different risk factors, for both sexes in 204 different countries and territories [7]. Data are updated every year, allowing us to properly compare all the measures mentioned from 1990 to 2019. No study to date has used the GBD dataset to quantify the burden of MSK conditions in the US.

Upon this basis, this study sought to characterize the burden of MSK disease in the US using the GBD data from 1990 to 2019. To the best of our knowledge, this is the very first paper to quantify the burden of MSK disorders in the US over time using GBD. A better understanding of this burden of disease is essential to treating and controlling the costs of these very prevalent and costly health conditions. The results of this study have the potential to help create interventions, inform policymaking, and lower healthcare costs in the US.

## 2. Methods

### 2.1. Data

For the investigation into MSK burden in the US, the current study utilized the 2019 iteration of the GBD dataset. This dataset offers comprehensive estimates encompassing various metrics such as deaths, incidence, prevalence, YLLs, YLDs, and DALYs attributable to a total of 369 diseases and injuries across 87 distinct risk factors. These estimates are available for both sexes and cover 204 different countries and territories, enabling a broad and detailed analysis [7,18]. Importantly, the data are regularly updated on an annual basis, facilitating temporal comparisons spanning from 1990 to 2019 for all the mentioned measures.

The GBD study is overseen by the Institute for Health Metrics and Evaluation (IHME) at the University of Washington in Seattle, with financial backing from diverse institutions including the World Bank, the National Institutes of Health, and the Bill & Melinda Gates Foundation [18,19].

Comprehensive datasets, featuring numerical values and age-standardized rates (ASRs) of prevalence and DALYs categorized by cause, sex, age, and geographical location, were procured from the Global Health Data Exchange website. DALYs were computed as the aggregate of YLLs, based on a reference maximum observed life expectancy, and YLDs utilizing standardized disability weights for each health state. The use of DALY rates proves especially informative for characterizing noncommunicable chronic diseases, given their distinctive impact characterized by low mortality and elevated disability rates.

### 2.2. Measures

We analyzed and presented trends in prevalence and DALYs rates spanning the period from 1990 to 2019 for individuals aged 5 years and older. Additionally, we conducted an in-depth exploration of the attributable risk factors and performed a decomposition analysis to understand the evolving trends in DALYs, aiming to identify and assess the underlying causes. Our graphical representations of the findings were generated using the R software package version 4.3.3 [20].

#### 2.2.1. DALYs

The burden of a specific disability is assessed using DALYs, a widely employed metric [21]. DALYs are computed by combining the total YLLs and YLDs, where YLLs are derived from a reference maximum life expectancy, and YLDs are determined using standardized disability weights for each health condition [18].

#### 2.2.2. Prevalence

Prevalence is a measure of the percentage of individuals in a population with a specific disease or condition at a given time. This metric, frequently utilized in epidemiology, serves to ascertain the disease burden within a population.

#### 2.2.3. MSK Disorders

We employed the American College of Rheumatology and the International Statistical Classification of Diseases and Related Health Problems (ICD-10) to delineate musculoskeletal disorders, as defined in the GBD dataset. The primary musculoskeletal disorders in the GBD dataset encompass the following: (1) low back pain; (2) occupationally related low back pain; (3) neck pain; (4) OA; (5) RA; (6) gout; (7) low bone mineral density; and (8) other MSK conditions. Under the ICD-10 classification, the category of other MSK conditions, as indicated in the GBD dataset, comprise an additional 13 diagnoses, including infectious arthropathies, inflammatory polyarthritis, and deforming dorsopathies.

#### 2.2.4. Risk Factors

The methodology employed for estimating the disease burden associated with risk factors in GBD 2019 has been comprehensively detailed elsewhere [18,22]. In alignment with this study’s objectives, all risk factors contributing to MSK disorders within the GBD 2019 dataset were encompassed. These include the following: (1) tobacco use (characterized as current daily or occasional use of any smoked tobacco product); (2) occupational risks (encompassing occupational injuries, ergonomic factors, and exposure to particulate matter, fumes, glasses, carcinogens, noise, and asthmagens); (3) kidney dysfunction (defined as estimated glomerular filtration rate (eGFR) less than 60 mL/min/1·73 m^2^ or albumin to creatinine ratio (ACR) greater than or equal to 30 mg/g); and (4) high body mass index (defined as Body Mass Index (BMI) greater than 25 kg/m^2^). The estimation of risk factor exposures involved utilizing population-representative survey and surveillance data, along with geospatial Gaussian process regression models that leveraged data strength across both time and geography.

### 2.3. Statistical Analysis

Briefly, the analysis methodologies have been comprehensively described in the existing literature [5,18,19,22,23,24,25,26]. The GBD study integrates data from various sources, applies covariate adjustments and modeling using standardized tools, and presents results through the GBD Compare website. Addressing missing data involves multiple imputation techniques for random missingness and alternative methods for non-random missing data, all while prioritizing data quality and transparency in documentation [3,22].

## 3. Results

### 3.1. Prevalence and DALYs by MSK Disorder and Sex

Table 1 shows age-standardized DALY rate, raw counts, and percentage change for different subcategories of MSK disorders in 1990 and 2019. In 2019, the age-standardized rate prevalence, DALY rates, and DALY numbers for all MSK disorders among both sexes was 30,161.0 [95% UI = 29,158.3–31,187.7], 4253.8 [95% UI = 3099.0–5558.5], and 13,951,637 [95% UI = 10,164,221–18,230,571], respectively. Between 1990 and 2019, the age-standardized rate for prevalence, DALY rates, and DALY numbers increased by 7.2%, 21.4%, and 57.0%, respectively. Low back pain continued to be the most prevalent MSK condition in 2019, exhibiting the highest contribution to age-standardized prevalence rates (42.1%), followed by other MSK disorders (35.6%), osteoarthritis (33.0%), and neck pain (17.0%).

Relatively different trends were observed when measuring the burden of disease, where low back pain was the highest contributor to DALY rates and numbers in 2019 (33.0% and 40.8%, respectively), followed by other MSK disorders (22.2% and 26.6%, respectively), neck pain (11.8% and 14.6%, respectively), and osteoarthritis (8.9% and 14.2%, respectively). Between 1990 and 2019, the largest percentage increase for DALY rates and numbers was witnessed for gout (85.8% and 204.4% increase, respectively) and other MSK disorders (55.9% and 121.9% increase, respectively). Despite the fact that lower back pain accounted for the highest burden of MSK disorders, it was the only condition that decreased in prevalence (−11.9%) between 1990 and 2019.

Table 2 shows the prevalence age-standardized rate, raw counts, and percentage change for different subcategories of MSK disorders in 1990 and 2019. In 2019, the age-standardized rate prevalence, DALY rates, and DALY numbers for all MSK disorders among males was 27,410.4 [95% UI = 26,472.7–28,357.1], 3584.5 [95% UI = 2592.7–4674.4], and 5,781,473 [95% UI = 4,181,685–7,539,334], respectively. Among females, the age-standardized rate prevalence, DALY rates, and DALY numbers for all MSK disorders were 32,817.5 [95% UI = 31,692.4–33,966.3], 4901.4 [95% UI = 3584.5–6424.7], and 8,170,165 [95% UI = 5,974,980–10,709,393], respectively. Between 1990 and 2019, the age-standardized rate for prevalence, DALY rates, and DALY numbers increased among males by 7.5%, 22.6%, and 59.7%, respectively. Among females, the age-standardized rate for prevalence, DALY rates, and DALY numbers increased by 7.5%. 20.8%, and 55.2%, respectively.

Furthermore, Table 3 shows age-standardized DALY rate, raw counts, and percentage change for different subcategories of MSK disorders per state and sex in 1990 and 2019. In terms of state-level variation, the District of Columbia had the lowest age-standardized DALYs rate attributed to MSK disorders in both sexes [2621.4 per 100,000; 95% UI = 1894.7–3453.2], followed by Hawaii [2738.6 per 100,000; 95% UI = 1992.9–3627.5] and Mississippi [2857.9 per 100,000; 95% UI = 2093.1–3712.7]. On the other hand, the three states with the highest age-standardized DALYs due to MSK disorders were North Dakota, Iowa, and South Dakota with rates of 3815.8 [95% UI = 2764.2–5.030.3], 3812.9 [95% UI = 2771.5–4990.1], and 3748.2 [95% UI = 2733.7–4918.0], respectively. While Pennsylvania witnessed the highest percentage change of 11.9% from 1990 to 2019, the District of Columbia and Hawaii were the only regions to witness a decrease with a percentage change of −2.9% and −0.4%, respectively.

### 3.2. Trends in DALYs, Prevalence, and Deaths for MSK Disorders (1990–2019)

Figure 1 shows the trends in DALYs, prevalence, and deaths for musculoskeletal disorders in the United States from 1990 to 2019. Between 1990 and 2019, there were changes in the trends in DALYs, prevalence, and deaths for MSK disorders. The prevalence rate per 100,000 started at 28,139.9 and underwent significant fluctuations, decreasing post-1990, followed by an increase post-1996, followed by a subsequent decline after 2001. Afterwards, there were minor oscillations over the years, ultimately exhibiting a substantial increase from 2006 to 2019. The DALYs rate per 100,000 exhibited subtle variations, both rising and falling over the years after starting from 3133.9 in 1990 but demonstrated a more pronounced increase particularly after 2010. The death rates per 100,000 exhibited minor changes, both increasing and decreasing after starting at 2.0 in 1990 and generally maintained a consistently low level.

### 3.3. Age-Specific Burden of Musculoskeletal Disorders (2019)

Figure 2 demonstrates DALYs for each musculoskeletal disorder across different age groups in the year 2019. By age group, ages 70–74, 75–79, and 65–69 had the highest age-standardized DALYs in 2019 while ages 5–9, 10–14, and 15–19 had the lowest age-standardized DALYs. The contribution of risk factors was similar across most age groups, with low back pain generally contributing the most risk, followed by OA, other MSK disorders, neck pain, and RA.

### 3.4. Geographical Distribution of DALY Rates

Figure 3 shows the age-standardized DALY rates for MSK disorders across 50 US states in 1990 and 2019. In 1990, the two highest DALY states were Iowa and then North Dakota, while the lowest DALY state was Mississippi. In 2019, the two highest DALY states were equally North Dakota and Iowa, while Mississippi remained the lowest DALY state.

### 3.5. Contribution of Risk Factors to DALYs

In 2019, high BMI, kidney dysfunction, occupational ergonomic factors, and smoking were the main risk factors for DALYs of MSK disorders. Figure 4 demonstrates the DALYs attributable to these four risk factors in 2019. High BMI appears to be the predominant risk factor across all age groups, with its impact peaking in the 50–69-year age range. The rate of DALYs associated with high body mass index progressively increases from the 15–19-year age group and demonstrates a gradual decline after the age of 70. Kidney dysfunction shows a more constant presence across age groups but becomes more significant as age increases, particularly after the age of 50. Occupational ergonomic factors are notable in the middle age groups, suggesting a potential link with active working years. Smoking, while less impactful than high body mass index, shows a substantial contribution to the rate of DALYs, with a notable increase from the age of 30 onwards.

### 3.6. Temporal Trends in Risk Factor Contributions (1990–2019)

In 2019, tobacco smoking, high BMI, and occupational risks, were the main risk factors for DALYs of MSK disorders, accounting for 284 per 100,000 [95% UI = 178–403], 241 per 100,000 [95% UI = 138–371], and 241 per 100,000 (95% UI = 169–319), respectively. Additionally, kidney dysfunction was a minor risk factor, contributing merely to 9 per 100,000 [95% UI = 6–12]. Between 1990 and 2019, there was a change in the trend of the highest risk factors contributing to age-standardized DALYs of MSK disorders, demonstrated in Figure 4. Between 1990 and 2019, tobacco smoking age-standardized DALYs contribution decreased from 428.9 per 100,000 [95% UI = 259–623] to 284 per 100,000 [95% UI = 178–403]. Despite this, tobacco smoking remained the largest risk factor for MSK disorders burden. On the other hand, the contribution of high BMI as a risk factor for MSK disorders increased from 188 per 100,000 age-standardized DALYs [95% UI = 99–310] to 241 per 100,000 [95% UI = 138–370]. Figure 5 demonstrates the change in age-standardized DALYs for each of the four risk factors between 1990 and 2019.

### 3.7. Sex-Specific Contribution of Risk Factors to DALYs

Figure 6 shows the contribution of the four main risk factors to DALYs of MSK disorders in females and males, respectively, in 2019. Both females and males showed a relatively similar trend of having smoking as the highest risk factor (37% and 36%, respectively) and kidney dysfunction as the lowest risk factor (1% and 2%, respectively). However, for females, the second highest risk factor after smoking was high body mass index (33%), followed by occupational ergonomic factors (29%). In contrast, for males the second highest risk factor after smoking was occupational ergonomic factors (33%), followed by high body mass index (29%).

## 4. Discussion

This study sought to investigate the burden of MSK disorders in the USA, using data from the GBD dataset from 1990 to 2019. To the best of our knowledge, our investigation is the first study to attempt to quantify the burden of MSK disorders in the USA using the GBD dataset, and one of the first studies to comprehensively investigate MSK disorders in the USA from an epidemiological perspective. Overall, upon analysis of the GBD dataset, the burden of MSK disorders remains very high from 1990 to 2019 indicating the need for effective intervention to promote better health outcomes.

The GBD dataset reveals a higher prevalence in the majority of MSK conditions in the female population. There were 485,593 affected females to 187,234 affected males with RA (prevalence ratio of 2.59). There were 13,272,059 affected females to 11,332,552 affected males with OA (prevalence ratio of 1.17). There were 10,399,283 affected females to 6,980,885 affected males with neck pain (prevalence ratio of 1.49). There were 15,803,242 affected females to 9,745,055 affected males with other MSK conditions (prevalence ratio of 1.62). The exceptions being gout and lower back pain, which had higher prevalence in males. There were 1,904,810 affected males to 550,791 affected females with gout, indicating men were 3.5 times more likely to experience gout and 22,962,287 affected males to 20,276,752 affected females with lower back pain indicating males were 1.1 times more likely to suffer from lower back pain. A closer look at the disparities in sex observed in the GBD dataset indicates that the burden of MSK conditions has a larger impact on the female population. The reasons for which come down to a multitude of physiological, anatomical, and psychosocial factors.

Females more commonly suffer from migraines, a condition that is linked with a 12 times higher prevalence rate of neck pain [27]. Conditions such as polymyalgia rheumatica and fibromyalgia are also more prevalent in the female population. Both conditions are commonly associated with neck pain [28]. Notably, females have an increased susceptibility to autoimmune diseases, including RA, due to sex-dimorphic immunological factors [29]. For instance, females exhibit heightened B-cell activity and a generally more responsive innate immune system, both of which contribute to autoimmune risks independent of hormonal influences. The increased estrogen to androgen ratio in females promotes a pro-inflammatory state that is believed to contribute to the progression of rheumatoid arthritis [30]. The peak of estrogen during pregnancy stimulates B cells and the Th2 response which results in the persistence of autoreactive B and T cell clones [30]. The decline in estrogen during menopause results in the release of pro-inflammatory cytokines IL-6, IL-1β, and TNF-α [30]. These events explain the high incidence of RA in females but particularly the increased incidence during pregnancy and menopause. Males have a higher articular cartilage thickness compared to females, which predisposes females to OA, particularly after menopause due to the lack of cartilage protective hormones [31]. Synovial fluid analysis in patients with OA reveals increased anti-inflammatory cytokines in males, and increased pro-inflammatory cytokines in females resulting in increased pain and inflammatory progression in females [32].

Recent studies into the effect of MSK conditions on the quality of life in female patients further elucidate the need for proper intervention to mitigate the burden of MSK conditions. It has been shown that RA has a negative impact on many aspects of sexual function including lubrication, orgasm, arousal, libido, and satisfaction [33,34].

Despite being the highest contributor to DALY rates and numbers in 2019, low back pain is the only MSK condition that has decreased in burden, decreasing by 14.2%. A rather perplexing finding given that the number of people suffering from low back pain continues to increase over time. Low back pain prevalence globally has increased from 377.5 million in 1990 to 557 million in 2017 with a projected 843 million in the year 2050 [35,36]. A potential explanation of this finding is that there is a great number of workers who underreport their pain and are less likely to call out of work to address their pain due to fear of job insecurity. A survey conducted by the Pew Research Center found that 73% of college educated Americans believe that there is currently less job security than 20–30 years ago [37]. Whether or not these claims are true, this illustrates a perception in the United States that job security is on the decline. Studies have shown that there is a significant association between job insecurity and not complaining of pain among younger people [38]. This trend could mask the true burden that low back pain has.

A closer look into the trends observed regarding age reveals an increase in the prevalence of MSK conditions in older populations from 1990 to 2019. With the highest DALYs seen in the age groups 65–69, 70–74, and 75–79. This trend can be explained in part by the growing population and prolonged life expectancy seen in the United States since 1990. The population has grown from 248.7 million to 336 million since 1990 to 2023, and the life expectancy has increased by approximately 5 years in that time period [39]. Overtime, the understanding of these conditions has improved and patients are more frequently visiting their healthcare providers [40], which could result in higher incidences of MSK conditions being diagnosed.

### 4.1. Risk Factors

Cigarette smoking has experienced a decline in recent decades. The CDC reports that the number of cigarette smokers has decreased from 20.9% to 11.5% from 2005 to 2021. This decline in cigarette smoking mirrors the downward trend seen in tobacco smoking age-standardized DALYs contribution from 1990 to 2019, decreasing from 428.9 per 100,000 to 284 per 100,000. However, with the substantial harm cigarette smoking causes, it is no surprise that it remains the largest risk factor for MSK disorders burden. Cigarette smoking has been shown to increase systemic inflammation and matrix metalloproteinase (MMP-12) [41]. MMP-12 is a macrophage elastase that has been implicated in the articular cartilage and joint destruction in RA [41]. This can exacerbate synovial inflammation in rheumatoid arthritis by impacting the same mechanisms that are already impaired in the disorder. Synovial fluid in RA contains higher levels of MMP-12 [41]. RA also has been shown to have decreased expression of proapoptotic factors FOXO1, RB1, TP53, and BAX which reduces the susceptibility of cell lines to apoptosis and increases inflammation and joint destruction [42]. The mechanism of injury due to cigarette smoking as well as the altered mechanisms in RA serve to illustrate the potential for synergistic harm that still keeps smoking as the number one risk factor for MSK disorders burden despite its decline in usage over the recent decades. Further studies may see this downward trend revert back and rise as the popularity of vaping continues to soar.

Obesity has been a cause of growing global concern over the last few decades. In the United States, the prevalence in obesity from 1994 to 2018 has increased from 23% to 43% [43]. This substantial rise in obesity parallels the upward trend discussed earlier in the contribution of high BMI as a risk factor for MSK disorders increasing from 188 per 100,000 age-standardized DALYs to 241 per 100,000. This trend can be attributed to the increased production of ultra processed foods, increased dependence on fast food services, and increased sedentary lifestyles over the recent decades. The NHANES survey shows that the American adult receives 57% of their energy intake from ultra processed foods [44]. With this continued rise in BMI, it is important to take a closer look into its impact on MSK disorders. Increased BMI has been shown to increase pain and stiffness via biomechanical stressors and promote inflammation [45]. This not only increases pain and lowers quality of life, but it also increases the progression of the MSK disorder. Increased BMI has been shown to increase the progression of osteoarthritis and increase the risk for total arthroplasty of the hip [46]. It is no surprise that along with the aforementioned trends, we also see that the largest percentage rise for DALY rates and numbers was seen with gout, jumping from 45.8% in 1990 to 66.7% in 2019. From 1990 to 2019, there has been an increase in the annual incidence of gout from 38.71 to 45.94 per 100,000 persons [47]. This rise in gout was observed in younger populations of varying socioeconomic backgrounds, with the largest rise seen in high income North American regions [47]. High BMI was found to be the greatest risk factor for years lived with disease for people with gout [47], further strengthening the link between the rise in obesity in the United States with the rise in gout. As the fastest growing risk factor for MSK disorders burden, it is imperative that swift and effective steps be taken to address the growing obesity rates in the United States.

Although there is a lack of definitive evidence to suggest that there has been a meaningful improvement in occupational policies to reduce MSK risks, more companies are becoming aware of the burden of MSK conditions and are exploring avenues to mitigate it. An example of such an endeavor can be seen in the Rolls Royce company where a study was conducted to determine the burden of MSK conditions on the company. It was found that 47% of the sick days accrued by employees within the study timeline were attributed to MSK conditions, costing the company GBP 50 million [48]. These employees were also found more likely to require mental health referrals and more days needed away from work [48]. This substantial loss in financial costs and in employee productivity will hopefully drive Rolls Royce and other companies to take initiatives at reducing occupation-related MSK risks. The US Bureau of Labor Statistics reports that healthcare and social assistance is the industry with the highest number of days away from work due to illness and injuries involving MSK disorders in 2018 with 56,360 days [49]. Followed by retail trade (41,070), manufacturing (38,640), and transportation and warehousing (38,350) [49]. The financial impact on all these different industries is generating more awareness. Perhaps this growing awareness and focus on minimizing MSK risks in the workplace that is more apparent today than in previous decades explains the modest decline seen in the age-standardized DALYs attributable to occupational risks from 1990 to 2019.

### 4.2. Public Health Implications

The findings of this study highlight critical public health implications, emphasizing the need for targeted interventions to address the identified challenges. The disproportionately high prevalence of MSK conditions in the female population underscores the importance of sex-specific interventions. Initiatives aimed at promoting awareness, early detection, and tailored treatment strategies for conditions such as rheumatoid arthritis, neck pain, migraines, polymyalgia rheumatica, and fibromyalgia could significantly improve outcomes for females [50]. Moreover, the paradoxical decrease in the burden of low back pain, despite its continued rise in prevalence, points to potential underreporting influenced by job insecurity perceptions. In this context, workplace-focused interventions, such as ergonomic improvements and mental health support, could mitigate the impact of job-related factors on the accurate reporting of MSK issues. Additionally, recognizing cigarette smoking and obesity as leading risk factors necessitates targeted public health campaigns to reduce smoking rates and address the growing obesity epidemic. These efforts can include anti-smoking initiatives, nutritional education, and initiatives promoting physical activity to alleviate biomechanical stressors and inflammation associated with MSK disorders. Furthermore, the study emphasizes the role of the workplace in MSK disorder prevention, exemplified by the Rolls Royce study [48]. Companies can implement measures like ergonomic workstations, health and wellness programs, and mental health support to reduce MSK-related sick days and enhance employee well-being. By focusing on these targeted interventions, public health initiatives can effectively address the multifaceted burden of MSK disorders in the United States.

### 4.3. Strengths and Limitations

Covering a span of three decades, our study provides invaluable insights into the long-term trends of musculoskeletal disorders in the US, highlighting changes and patterns that are crucial for understanding the evolution of these conditions. A key strength of our research lies in the use of the latest 2019 version of the GBD dataset, which has undergone significant enhancements in terms of data scope, quality, and comprehensiveness, especially in previously underrepresented regions. These improvements have enabled a more accurate and thorough analysis [19]. In terms of methodology, our study incorporates the latest scientific evidence and advanced disease classifications, along with refined uncertainty estimates, ensuring that our analysis is both current and robust. We have also taken a deep dive into various risk factors associated with musculoskeletal disorders, such as high BMI, kidney dysfunction, occupational ergonomic factors, and smoking. This not only aids in understanding the multifaceted nature of these disorders but also provides critical insights for potential preventive strategies. Furthermore, our research marks a pioneering effort in using the GBD 2019 data for an epidemiological analysis of musculoskeletal disorders in the United States, filling a significant gap in existing research. By providing quantifiable measures and thoroughly examining an understudied area, our study contributes valuable evidence-based directions for public health interventions and policymaking. Additionally, our findings underscore the urgent need for enhanced focus on rehabilitation medicine as a specialty in the US healthcare system, thus presenting a compelling call to action for healthcare providers and policymakers alike.

While our study offers significant insights into musculoskeletal disorders in the United States over three decades, it is important to recognize certain limitations. A primary limitation is the reliance on secondary data from the Global Burden of Disease study. Despite GBD’s rigorous methods and estimations, secondary data can sometimes be less precise than primary data collection, potentially leading to inaccuracies or incomplete representations [51].

Particularly in the US context, there might be disparities in data quality and availability, especially in certain demographic groups or specific geographic areas. This could result in potential errors or biases in our estimates, possibly leading to underestimation or overestimation of the true burden of musculoskeletal disorders. Additionally, while our study effectively identifies associations between various risk factors and musculoskeletal disorders, it does not establish causation. The ecological nature of the study means that our findings might be subject to ecological fallacies, and the lack of individual-level data restricts our capacity to make conclusive statements about individual risk.

Another limitation is the categorization of some musculoskeletal conditions. For example, different forms of low back pain, which can vary significantly in their etiology and clinical implications, were not always distinctly classified. This aggregation might have limited our ability to offer more precise findings and targeted intervention strategies [40]. We also recognize that there could be significant confounders and variables, such as lifestyle factors, comorbidities, access to healthcare, and treatment modalities, that interact with the DALYs associated with musculoskeletal disorders. Although these limitations are inherent to the GBD study methodology and were beyond our control, they must be acknowledged in the interpretation of our results.

### 4.4. Future Research

Future research should aim to build on the insights gained from the last three decades, while also exploring new methodologies and interdisciplinary approaches. One promising avenue is the expansion of comparative studies that juxtapose the trends observed in the US with those in other regions globally. This comparative analysis could reveal region-specific factors influencing the prevalence and management of MSK disorders, offering a global perspective that could inform more effective prevention and treatment strategies. Additionally, there is a significant need to validate our findings through the utilization of diverse and expansive data sources. National health surveys, hospital records, insurance claims data, and longitudinal studies would not only corroborate the trends identified but also provide a more nuanced understanding of the progression and causation of MSK disorders. Such studies are particularly crucial in establishing definitive links between various risk factors, such as socioeconomic status, lifestyle choices, and genetic predispositions, and the incidence and severity of MSK disorders.

In the context of healthcare utilization and outcomes, future research must delve into how different treatment protocols and rehabilitation strategies impact patient outcomes in the US. This involves analyzing healthcare utilization patterns and their efficacy, offering insights into optimizing resource allocation and improving patient care. The role of technology, especially digital health solutions like telehealth and mobile health applications, should also be a focus area. These technologies have the potential to revolutionize the management of MSK disorders, particularly in a diverse and geographically vast country like the US. Furthermore, understanding the effectiveness of public health campaigns and prevention strategies in the US could provide valuable lessons in reducing the burden of these disorders. Such research would benefit greatly from an interdisciplinary approach, integrating perspectives from epidemiology, genetics, public health, and health policy.

## 5. Conclusions

In conclusion, our comprehensive study on the burden of MSK disorders in the United States over a span of three decades using the GBD dataset has revealed significant insights. It highlighted the persistent high prevalence of MSK disorders, with a notable impact on the female population and a complex interplay of physiological, behavioral, and environmental risk factors. Our findings exemplify the need for sex-specific interventions, workplace-focused strategies, and public health campaigns addressing key risk factors like smoking and obesity. By offering a detailed epidemiological analysis and identifying trends and disparities, this study provides a crucial foundation for informed public health interventions and policymaking, especially in the realm of MSK disorders. Future research should focus on comparative global studies, comprehensive data analysis from diverse sources, and the exploration of novel healthcare technologies and strategies. This will not only enrich our understanding of MSK disorders but also guide the development of more effective prevention and treatment approaches, ultimately improving health outcomes for affected populations.

## Figures and Tables

**Figure 1 jcm-13-06732-f001:**
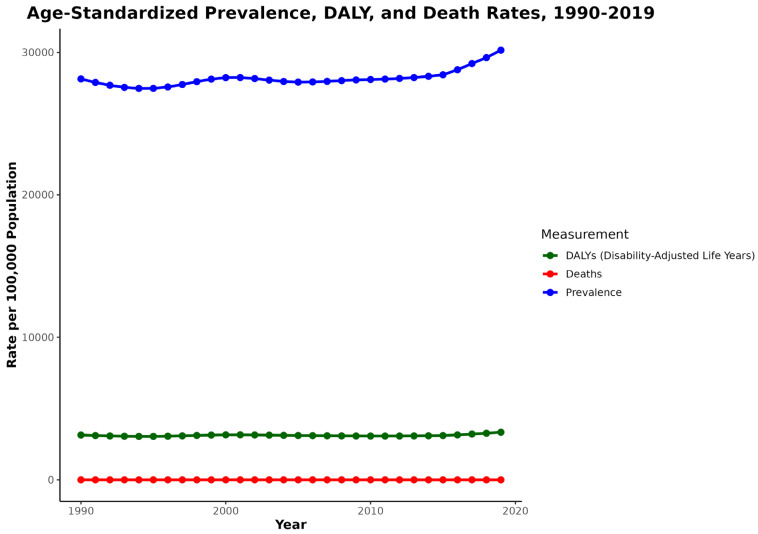
Trends of age-standardized prevalence, DALY, and death rates from 1990 to 2019.

**Figure 2 jcm-13-06732-f002:**
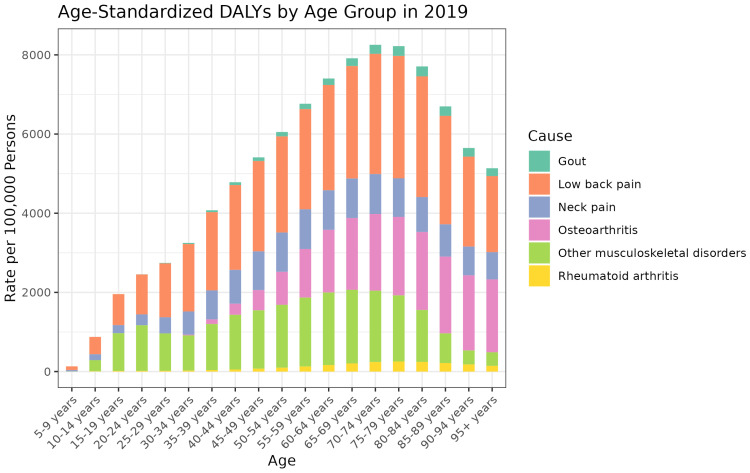
Age-standardized DALY rates by cause and age group in 2019.

**Figure 3 jcm-13-06732-f003:**
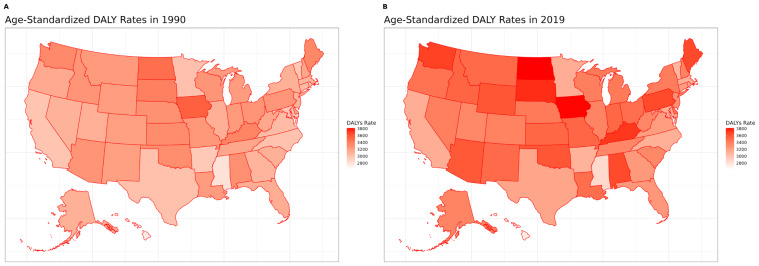
Age-standardized DALY rates by state, 1990 vs. 2019.

**Figure 4 jcm-13-06732-f004:**
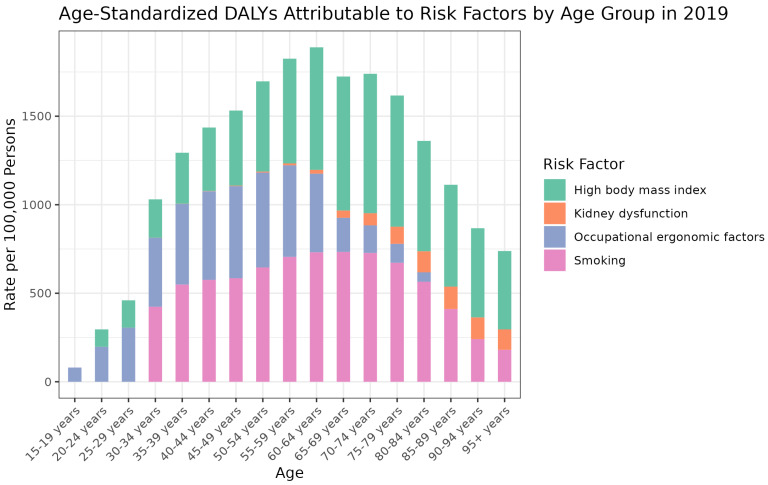
Age-standardized DALYs rate attributable to risk factors by age group in 2019.

**Figure 5 jcm-13-06732-f005:**
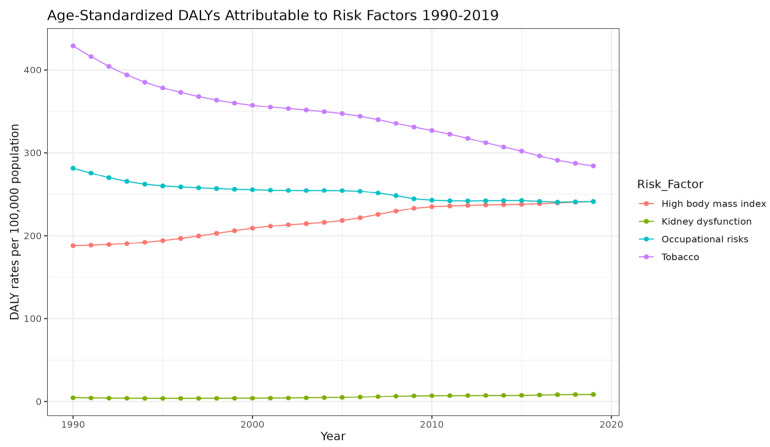
Trends of risk factors attributing to age-standardized DALYs from 1990 to 2019.

**Figure 6 jcm-13-06732-f006:**
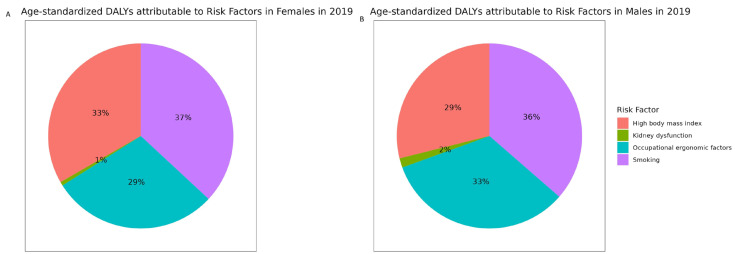
Age-standardized DALYs attributable to specific risk factors in females (**A**) and males (**B**) in 2019.

**Table 1 jcm-13-06732-t001:** DALY age-standardized rate, raw counts, and percentage change for different subcategories of MSK disorders in 1990 and 2019. Positive values indicate increase while negative values indicate decrease.

Cause	Sex	Age-Standardized Rate 1990 (95% CI)	Age-Standardized Rate 2019 (95% CI)	Percentage Change (%)	Raw DALY Counts in 1990 (95% CI)	Raw DALY Counts in 2019 (95% CI)	Percentage Change (%)
Musculoskeletal disorders	Male	2923.2 (2108.4–3872.1)	3584.5 (2592.7–4674.4)	22.6%	3,619,373 (2,610,494–4,794,206)	5,781,473 (4,181,685–7,539,334)	59.7%
	Female	4056.2 (2951.9–5323.4)	4901.4 (3584.5–6424.7)	20.8%	5,264,586 (3,831,275–6,909,270)	8,170,165 (5,974,980–10,709,393)	55.2%
	Both	3503.1 (2545.4–4619.3)	4253.8 (3099.0–5558.5)	21.4%	8,883,959 (6,455,298–11,714,709)	13,951,637 (10,164,221–18,230,571)	57.0%
Gout	Male	47.3 (29.7–67.4)	90.2 (59.5–125.5)	90.7%	63,353 (39,684–89,245)	203,403 (133,157–282,962)	221.1%
	Female	13.0 (8.3–18.5)	18.6 (12.2–25.6)	43.1%	22,712 (14,504–32,206)	55,020 (36,429–75,754)	142.3%
	Both	28.7 (18.1–40.6)	53.0 (35.0–73.6)	84.7%	86,066 (54,878–121,733)	258,423 (171,072–356,153)	200.3%
Low back pain	Male	1473.6 (1041.2–1978.7)	1287.1 (921.3–1704.6)	−12.7%	1,968,132 (1,386,474–2,644,345)	2,528,568 (1,824,739–3,343,901)	28.5%
	Female	1721.8 (1214.1–2309.5)	1515.4 (1090.0–1998.0)	−12.0%	2,536,733 (1,788,828–3,395,652)	3,168,584 (2,289,539–4,132,010)	24.9%
	Both	1602.7 (1131.7–2146.6)	1402.9 (1007.0–1852.3)	−12.5%	4,504,864 (3,168,679–6,039,640)	5,697,152 (4,114,139–7,474,690)	26.5%
Neck pain	Male	360.7 (239.1–515.5)	407.0 (278.3–578.2)	12.8%	483,165 (321,197–687,678)	814,748 (552,367–1,148,870)	68.6%
	Female	481.3 (320.9–689.6)	591.4 (400.4–833.4)	22.9%	718,457 (475,228–1,018,020)	1,228,771 (839,592–1,732,617)	71.0%
	Both	422.5 (279.7–604.6)	500.3 (338.9–704.9)	18.4%	1,201,621 (792,529–1,709,086)	2,043,518 (1,392,665–2,886,397)	70.1%
Osteoarthritis	Male	301.0 (152.1–608.6)	298.5 (152.3–606.4)	−0.8%	399,549 (202,606–810,305)	729,300 (373,054–1,471,103)	82.5%
	Female	411.9 (210.0–826.9)	449.3 (230.0–896.9)	9.1%	688,430 (354,031–1,388,126)	1,257,043 (648,596–2,509,014)	82.6%
	Both	361.6 (183.7–726.9)	378.1 (194.8–759.8)	4.6%	1,087,980 (556,601–2,194,304)	1,986,343 (1,023,960–3,964,292)	82.6%
Rheumatoid arthritis	Male	27.4 (20.7–34.5)	30.0 (22.7–38.6)	9.5%	36,856 (27,824–46,272)	69,981 (52,903–89,594)	89.9%
	Female	65.4 (48.6–83.4)	75.2 (55.5–96.3)	15.0%	105,882 (78,707–135,415)	187,902 (138,400–239,464)	77.5%
	Both	47.8 (35.9–60.5)	53.5 (39.6–68.3)	11.9%	142,738 (107,623–180,111)	257,884 (192,870–328,515)	80.7%
Other musculoskeletal disorders	Male	501.9 (345.0–692.2)	763.3 (544.3–1031.3)	52.1%	668,317 (460,403–923,656)	1,435,473 (1,027,490–1,927,794)	114.8%
	Female	824.2 (590.6–1116.1)	1114.1 (802.4–1480.7)	35.2%	1,192,372 (852,371–1,622,073)	2,272,845 (1,633,281–2,992,696)	90.6%
	Both	670.6 (480.6–912.8)	942.8 (677.6–1259.8)	40.6%	1,860,689 (1,325,225–2,545,320)	3,708,318 (2,691,090–4,914,367)	99.3%

**Table 2 jcm-13-06732-t002:** Prevalence age-standardized rate, raw counts, and percentage change for different subcategories of MSK disorders in 1990 and 2019.

Cause	Sex	Age-Standardized Rate 1990 (95% CI)	Age-Standardized Rate 2019 (95% CI)	Percentage Change (%)	Raw Prevalence Value in 1990 (95% CI)	Raw Prevalence Value in 2019 (95% CI)	Percentage Change (%)
Musculoskeletal disorders	Male	25,491.0 (24,125.2–26,900.4)	27,410.4 (26,472.7–28,357.1)	7.5%	33,943,087 (32,117,243–35,825,521)	55,740,384 (53,938,569–57,628,455)	64.2%
	Female	30,597.5 (29,093.2–32,192.7)	32,817.5 (31,692.4–33,966.3)	7.3%	46,105,639 (43,895,515–48,305,417)	71,670,741 (69,331,000–74,036,455)	55.4%
	Both	28,139.9 (26,717.3–29,630.5)	30,161.0 (29,158.3–31,187.7)	7.2%	80,048,727 (76,125,913–84,127,340)	127,411,125 (123,354,438–131,646,893)	59.2%
Gout	Male	1554.2 (1225.2–1936.6)	2962.3 (2535.0–3395.3)	90.6%	2,078,067 (1,628,445–2,594,343)	6,743,500 (5,746,946–7,854,449)	224.5%
	Female	432.6 (343.3–540.5)	636.2 (539.9–749.0)	47.1%	765,615 (603,786–956,608)	1,912,067 (1,608,975–2,249,279)	149.7%
	Both	942.7 (746.7–1176.9)	1752.0 (1507.1–2016.7)	85.8%	2,843,682 (226,4491–3,526,983)	8,655,567 (7,420,981–10,035,563)	204.4%
Low back pain	Male	13,148.4 (11,668.5–14,938.1)	11,557.6 (10,665.8–12,510.1)	−12.1%	17,542,901 (1,556,0381–19,890,446)	22,878,611 (21,138,532–24,893,804)	30.4%
	Female	15,584.4 (13,804.5–17,607.2)	13,818.4 (12,718.3–14,943.1)	−11.3%	23,093,509 (20,497,084–25,985,376)	29,226,817 (27,041,081–31,650,286)	26.6%
	Both	14,414.9 (12,814.4–16,299.9)	12,706.0 (11,718.1–13,778.9)	−11.9%	40,636,410 (36,177,564–45,823,737)	52,105,428 (48,196,029–56,404,010)	28.2%
Neck pain	Male	3675.4 (2939.1–4606.0)	4145.1 (3439.8–4972.2)	12.8%	4,922,450 (3,940,986–6,155,959)	8,385,528 (6,928,142–10,043,800)	70.4%
	Female	4943.6 (3971.3–6284.0)	6077.7 (5039.6–7365.9)	22.9%	7,425,648 (6,023,089–9,313,174)	12,798,821 (10,650,652–15,327,201)	72.4%
	Both	4325.4 (3473.7–5461.4)	5123.3 (4268.4–6170.4)	18.4%	12,348,099 (9,975,795–15,461,040)	21,184,349 (17,566,737–25,306,220)	71.6%
Osteoarthritis	Male	8258.8 (7396.9–9309.6)	8208.9 (7434.5–9154.5)	−0.6%	10,945,968 (9,818,649–12,319,128)	19,950,890 (18,003,623–22,330,817)	82.3%
	Female	10,806.5 (9653.9–12,113.5)	11,544.6 (10,401.5–12,875.5)	6.8%	17,927,241 (16,092,636–19,945,386)	31,914,999 (28,799,235–35,529,128)	78.0%
	Both	9640.8 (8608.6–10,797.1)	9960.9 (9003.8–11,110.5)	3.3%	28,873,209 (25,939,261–32,228,469)	51,865,889 (46,804,272–57,860,984)	79.6%
Rheumatoid arthritis	Male	159.1 (150.0–169.2)	184.5 (172.1–197.5)	16.0%	212,951 (200,832–226,398)	425,600 (395,648–456,233)	99.9%
	Female	410.0 (388.2–435.0)	490.4 (460.5–525.1)	19.6%	649,184 (615,255–687,174)	1,197,173 (1,119,718–1,280,381)	84.4%
	Both	292.4 (277.1–310.3)	342.6 (321.7–366.1)	17.2%	862,136 (817,097–913,209)	1,622,773 (1,513,606–1,734,740)	88.2%
Other musculoskeletal disorders	Male	5595.9 (4709.3–6591.3)	8724.9 (8176.2–9342.6)	55.9%	7,458,138 (6,255,541–8,763,627)	16,546,635 (15,414,657–17,797,363)	121.9%
	Female	9077.6 (7937.5–10,373.8)	12,637.8 (11,868.8–13,473.3)	39.2%	13,149,980 (11,458,125–14,999,864)	25,996,108 (24,142,665–27,974,417)	97.7%
	Both	7420.7 (6429.8–8574.0)	10,727.3 (10,069.3–11,442.6)	44.6%	20,608,118 (17,774,822–23,757,801)	42,542,743 (39,663,782–45,681,861)	106.4%

**Table 3 jcm-13-06732-t003:** DALY age-standardized rate and percentage change for all MSK disorders in 1990 and 2019 across states. Positive values represent increase in percent change while negative values indicate decrease.

State	Sex	Age-Standardized DALYs Rate in 1990	Age-Standardized DALYs Rate in 2019	% Change
Alabama	Male	2828.5 (2029.2–3742.3)	3157.6 (2282.7–4145.9)	11.6%
	Female	3716.5 (2717.1–4897)	4102.7 (3000.6–5313)	10.4%
	Both	3301.7 (2396.8–4353)	3647.2 (2658.4–4745.4)	10.5%
Alaska	Male	2676.9 (1925.7–3565.7)	2889.4 (2081.5–3785.2)	7.9%
	Female	3574.1 (2582–4727.1)	3892.3 (2825.7–5078.7)	8.9%
	Both	3104.1 (2242.3–4104.3)	3366.3 (2443.9–4419.5)	8.4%
Arizona	Male	2854.8 (2066.2–3769.7)	3119.9 (2255.3–4086.8)	9.3%
	Female	3671.5 (2682.9–4837.8)	4034.2 (2962.7–5269.1)	9.9%
	Both	3276.7 (2384.4–4337.4)	3587.1 (2619.3–4683.3)	9.5%
Arkansas	Male	2528.3 (1808.4–3361)	2638.2 (1902.9–3453.4)	4.3%
	Female	3340.6 (2412.8–4407.2)	3530.5 (2569.5–4604.7)	5.7%
	Both	2956 (2127.4–3914)	3093.9 (2250.6–4037.4)	4.7%
California	Male	2582.4 (1863.5–3402.9)	2704.7 (1937–3546.9)	4.7%
	Female	3364.7 (2446.8–4474.7)	3525.6 (2572.3–4601.5)	4.8%
	Both	2983 (2161.6–3958.1)	3122.4 (2257.9–4086.4)	4.7%
Colorado	Male	2759 (1993.6–3647.2)	2899.4 (2094.1–3794.3)	5.1%
	Female	3632.6 (2652.4–4768.1)	3848.1 (2817.7–5009.2)	5.9%
	Both	3209 (2334.2–4213.5)	3378.9 (2459.9–4396.4)	5.3%
Connecticut	Male	2634.6 (1882.8–3508.4)	2770.6 (2014.8–3646.9)	5.2%
	Female	3412.4 (2448.1–4502.1)	3643 (2641.4–4806.9)	6.8%
	Both	3042.1 (2192.3–4007.3)	3218.7 (2334.2–4227.8)	5.8%
Delaware	Male	2782.8 (1987.4–3694.8)	2989 (2165.9–3916.5)	7.4%
	Female	3582.2 (2597.2–4693.3)	3905.2 (2846.8–5117)	9.0%
	Both	3202.7 (2308.8–4209.5)	3466.9 (2517.1–4542.7)	8.2%
District of Columbia	Male	2286.4 (1631.4–3025.4)	2205.6 (1590.1–2908.7)	−3.5%
	Female	3043.5 (2212.4–3998.9)	2997.9 (2183.4–3936.3)	−1.5%
	Both	2700 (1961.6–3554)	2621.4 (1894.7–3453.2)	−2.9%
Florida	Male	2729.8 (1947.7–3616.6)	2825.9 (2044.2–3701)	3.5%
	Female	3495.9 (2532.9–4565.3)	3655.2 (2675.4–4758.9)	4.6%
	Both	3131.2 (2260.1–4105.8)	3252.8 (2375.4–4235.2)	3.9%
Georgia	Male	2676.9 (1926.7–3552.9)	2833.4 (2044.9–3703.7)	5.8%
	Female	3440.4 (2489.2–4526.6)	3627.9 (2645.4–4715.4)	5.4%
	Both	3081.7 (2221.9–4059)	3249.2 (2361–4216)	5.4%
Hawaii	Male	2436.4 (1739.1–3233.9)	2454.2 (1768.4–3260.7)	0.7%
	Female	3061.9 (2209.9–4039.3)	3022.2 (2200.3–3987.6)	−1.3%
	Both	2749.7 (1977.9–3646.9)	2738.6 (1992.9–3627.5)	−0.4%
Idaho	Male	2845.1 (2052.4–3762.3)	3065.8 (2229.4–4022)	7.8%
	Female	3688.7 (2691.4–4877.4)	4000 (2948.7–5193.6)	8.4%
	Both	3273.9 (2372.3–4318.7)	3535 (2601–4608.6)	8.0%
Illinois	Male	2691.5 (1925.8–3576)	2900.4 (2090.3–3808)	7.8%
	Female	3503.8 (2533.7–4616.3)	3811.8 (2781.7–5001.7)	8.8%
	Both	3118.8 (2246–4124.2)	3367.3 (2440.2–4420.7)	8.0%
Indiana	Male	2797.8 (2004.3–3704.5)	3023.9 (2183.2–3944.4)	8.1%
	Female	3639.7 (2637.3–4792.9)	3996.9 (2925.7–5194.9)	9.8%
	Both	3240.7 (2341.8–4275.3)	3521.5 (2556.5–4597.7)	8.7%
Iowa	Male	3119.5 (2247.5–4103.9)	3292.5 (2393.2–4325.8)	5.5%
	Female	3949.4 (2869.6–5174.4)	4332.6 (3152.1–5643.5)	9.7%
	Both	3551.6 (2579.3–4656.2)	3812.9 (2771.5–4990.1)	7.4%
Kansas	Male	2870.6 (2068.4–3815.5)	3023.1 (2176.6–3991.6)	5.3%
	Female	3741.7 (2712.4–4873.4)	4012.3 (2939.6–5216.4)	7.2%
	Both	3321.3 (2400.2–4367)	3521 (2567.6–4614.1)	6.0%
Kentucky	Male	2919.1 (2108.9–3859.6)	3242.5 (2342.4–4260.7)	11.1%
	Female	3759.8 (2726.6–4943)	4178.8 (3070.7–5417)	11.1%
	Both	3361.9 (2432.1–4428.2)	3718.6 (2700.6–4841.4)	10.6%
Louisiana	Male	2796.6 (2018–3711.7)	3019.9 (2178.8–3938.6)	8.0%
	Female	3616.1 (2638.9–4749)	3924.8 (2890.2–5111.1)	8.5%
	Both	3233.8 (2343.5–4262.7)	3483.7 (2542.8–4525.1)	7.7%
Maine	Male	2891.8 (2072–3813.9)	3079.4 (2235.9–4041.9)	6.5%
	Female	3758.4 (2720.6–4943.1)	4216.5 (3091.5–5448.2)	12.2%
	Both	3343.3 (2406.9–4400.2)	3654.6 (2661.6–4775.4)	9.3%
Maryland	Male	2552 (1832.3–3414.8)	2642.8 (1901.5–3462.7)	3.6%
	Female	3364.6 (2415.3–4416.8)	3560.5 (2589.5–4657.3)	5.8%
	Both	2980.8 (2151.8–3957.8)	3121 (2249.8–4072.8)	4.7%
Massachusetts	Male	2690.5 (1915.7–3558.7)	2702.7 (1954–3538.9)	0.5%
	Female	3441.1 (2488.5–4532.9)	3498 (2536.2–4566.2)	1.7%
	Both	3088.8 (2233.7–4074.8)	3114.9 (2255.7–4052.2)	0.8%
Michigan	Male	2806.9 (2017.1–3707.6)	2950.5 (2152.6–3872.2)	5.1%
	Female	3609.5 (2605.8–4741.8)	3867.4 (2824.3–5044.4)	7.1%
	Both	3229.2 (2340.8–4245.2)	3420.6 (2488.1–4469)	5.9%
Minnesota	Male	2550.4 (1830.2–3407.5)	2617.2 (1896.6–3454.9)	2.6%
	Female	3432.4 (2492.2–4515.5)	3632.7 (2663.5–4750.1)	5.8%
	Both	3006.6 (2175–3967.6)	3128.1 (2265–4109.6)	4.0%
Mississippi	Male	2445.8 (1749.1–3245.9)	2524.7 (1820.1–3305.8)	3.2%
	Female	3115.4 (2258.8–4116.8)	3165.6 (2310.2–4114.4)	1.6%
	Both	2804.6 (2027.3–3713.5)	2857.9 (2093.1–3712.7)	1.9%
Missouri	Male	2800.5 (2008.8–3711.3)	3015.4 (2189.1–3948.9)	7.7%
	Female	3670.5 (2658.4–4835.4)	4013.6 (2932.5–5236.6)	9.3%
	Both	3259.8 (2360.4–4291.4)	3524.9 (2567.4–4619.6)	8.1%
Montana	Male	2742 (1973.7–3656.3)	2921.9 (2112.6–3829.6)	6.6%
	Female	3719.2 (2692.1–4859.9)	4093.6 (3021.5–5339.3)	10.1%
	Both	3239.6 (2341.1–4283.5)	3501.8 (2572.8–4565.6)	8.1%
Nebraska	Male	2780.7 (1994.8–3697)	2892.7 (2093.4–3817.4)	4.0%
	Female	3763.7 (2725.8–4971.3)	3946.6 (2887.2–5170.3)	4.9%
	Both	3291.3 (2380.3–4369.8)	3422.1 (2487.5–4498.2)	4.0%
Nevada	Male	2698.9 (1934.6–3586.9)	2971 (2137.5–3885)	10.1%
	Female	3502.4 (2517.9–4597.3)	3811 (2761.8–4949.6)	8.8%
	Both	3097.4 (2224.7–4089.4)	3393.2 (2460.1–4430.3)	9.5%
New Hampshire	Male	2702.1 (1939–3580.5)	2851.3 (2059.4–3733.5)	5.5%
	Female	3599.7 (2610.5–4745.1)	3880.7 (2841.9–5050.8)	7.8%
	Both	3168.1 (2292.2–4182.2)	3371.9 (2462.4–4399.8)	6.4%
New Jersey	Male	2705.3 (1934.3–3602.2)	2894.5 (2096.3–3803)	7.0%
	Female	3443.1 (2507–4536.1)	3680 (2697–4789.4)	6.9%
	Both	3094.5 (2235.1–4070.9)	3300 (2407.2–4310.6)	6.6%
New Mexico	Male	2702.6 (1948.6–3579.4)	2920.4 (2116.7–3819.7)	8.1%
	Female	3679.2 (2675.4–4830.5)	4061.6 (2993.9–5273)	10.4%
	Both	3208.4 (2325.8–4219.5)	3498.7 (2565.4–4552.8)	9.0%
New York	Male	2706.8 (1955.1–3593.7)	2978.8 (2147.5–3875.3)	10.0%
	Female	3451.4 (2478.2–4552.9)	3797.9 (2758.2–4951.5)	10.0%
	Both	3104 (2235.9–4110.8)	3404.5 (2468.9–4419.3)	9.7%
North Carolina	Male	2592.7 (1869.2–3435.3)	2689.4 (1940.3–3531.1)	3.7%
	Female	3366.7 (2463.8–4411.1)	3496 (2562.3–4582.8)	3.8%
	Both	3001.9 (2187–3959.4)	3109.7 (2262.8–4081.2)	3.6%
North Dakota	Male	2906.8 (2087.4–3842.5)	3206.1 (2314.6–4222.2)	10.3%
	Female	4024.7 (2878.9–5392.1)	4449.8 (3214.9–5894.8)	10.6%
	Both	3478.2 (2496.6–4632.4)	3815.8 (2764.2–5030.3)	9.7%
Ohio	Male	2835.3 (2049.4–3770.7)	3001.9 (2166.7–3927.2)	5.9%
	Female	3668.2 (2639.6–4828.9)	3942.8 (2868.3–5129.6)	7.5%
	Both	3276.6 (2357–4340.3)	3484.7 (2528.6–4542)	6.4%
Oklahoma	Male	2848.7 (2027.2–3782.6)	3096.5 (2246.4–4073.6)	8.7%
	Female	3728.7 (2693.6–4907)	4091.1 (2995.5–5308.4)	9.7%
	Both	3308.9 (2378.2–4367.8)	3599.4 (2618–4713.4)	8.8%
Oregon	Male	2682.2 (1929.4–3549.3)	2898.6 (2095.3–3816.6)	8.1%
	Female	3585 (2612.2–4751)	3999.5 (2923.8–5206.8)	11.6%
	Both	3146.8 (2275.9–4152)	3457.4 (2517.4–4523.1)	9.9%
Pennsylvania	Male	2869.3 (2049.3–3789.8)	3209.7 (2330.6–4223.6)	11.9%
	Female	3609.3 (2627.1–4762.3)	4075.9 (2983–5303.4)	12.9%
	Both	3262 (2359.2–4299)	3651.6 (2658–4782.3)	11.9%
Rhode Island	Male	2543.2 (1817.9–3354.1)	2654.4 (1912.2–3485.1)	4.4%
	Female	3415.3 (2458.9–4501.7)	3612.4 (2625.2–4703.6)	5.8%
	Both	3005.1 (2167.4–3970.6)	3149.3 (2279.5–4113.8)	4.8%
South Carolina	Male	2682 (1928.6–3532.3)	2897.1 (2094.2–3802.1)	8.0%
	Female	3479.1 (2539.3–4604)	3768.4 (2759.1–4916.2)	8.3%
	Both	3105 (2252.5–4110.9)	3347 (2451.9–4376.3)	7.8%
South Dakota	Male	2782.2 (2003.6–3706.2)	3097.4 (2226.6–4058.7)	11.3%
	Female	3962 (2902.9–5210.4)	4411.6 (3245.2–5760.5)	11.3%
	Both	3386.5 (2471.3–4468.8)	3748.2 (2733.7–4918)	10.7%
Tennessee	Male	2745.1 (1982.3–3636.4)	2907 (2107.3–3781.8)	5.9%
	Female	3539.7 (2575.5–4637.7)	3773.3 (2779.5–4938.8)	6.6%
	Both	3165.5 (2301.9–4166.4)	3355.4 (2458.5–4406.2)	6.0%
Texas	Male	2584.3 (1854–3435.4)	2713.3 (1960.9–3580.3)	5.0%
	Female	3398.1 (2469.7–4475.8)	3615.5 (2633.7–4720.8)	6.4%
	Both	3008.2 (2177.9–3988.5)	3174.9 (2307.8–4153.6)	5.5%
Utah	Male	2670.6 (1919–3536)	2859.1 (2072.7–3761.9)	7.1%
	Female	3528.5 (2575.4–4649.9)	3815.9 (2793.6–4987.2)	8.1%
	Both	3113.7 (2258.5–4106.8)	3342.4 (2425.6–4375.4)	7.3%
Vermont	Male	2574.4 (1841.3–3432.4)	2607.7 (1888.9–3441.3)	1.3%
	Female	3329 (2412–4396)	3519.6 (2560.2–4597.5)	5.7%
	Both	2965.3 (2145.1–3928.9)	3068.3 (2234.1–4002.3)	3.5%
Virginia	Male	2623.9 (1880.8–3464.4)	2758.5 (1985.5–3636.3)	5.1%
	Female	3427.1 (2470.5–4498.4)	3659.7 (2667.7–4772.7)	6.8%
	Both	3044.3 (2190.7–4001.3)	3221.2 (2346.3–4208.8)	5.8%
Washington	Male	2864.8 (2058.4–3789.6)	3166.2 (2296.5–4154.4)	10.5%
	Female	3778.3 (2719.9–5020.1)	4192.4 (3055.4–5497.2)	11.0%
	Both	3332.5 (2403.4–4419.4)	3683.2 (2676.5–4825.6)	10.5%
West Virginia	Male	2708.1 (1942.3–3606.5)	2851.3 (2066.9–3744.4)	5.3%
	Female	3559.2 (2591.9–4717.5)	3846.7 (2814.5–4981.4)	8.1%
	Both	3158.6 (2286–4195.5)	3351.8 (2441–4366.3)	6.1%
Wisconsin	Male	2838.3 (2037.5–3758.9)	2928.9 (2133.4–3856.6)	3.2%
	Female	3671.1 (2654.7–4836.5)	3869 (2819.3–5057.7)	5.4%
	Both	3269.5 (2365.8–4321.8)	3400.7 (2476.5–4436.4)	4.0%
Wyoming	Male	2867.9 (2058.5–3799.9)	3197.9 (2316.1–4205.2)	11.5%
	Female	3571.5 (2606.3–4705)	4024.6 (2941.3–5283.5)	12.7%
	Both	3224.3 (2338.3–4258)	3598.1 (2615.5–4725.2)	11.6%

## Data Availability

All data used in this research work are publicly available through the GBD dataset https://vizhub.healthdata.org/gbd-results/ (accessed on 4 November 2024).

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
