# Peer review of "Quantifying the Strain: A Global Burden of Disease (GBD) Perspective on Musculoskeletal Disorders in the United States Over Three Decades: 1990–2019"

_jcm, 2024, doi:10.3390/jcm13226732_

Round 1

Reviewer 1 Report

Comments and Suggestions for Authors

Dear Authors,

Interesting topic and, as you said - one of the first studies to describe the epidemiological aspect of musculoskeletal disorders in the USA. Introduction widely describes the entity and importance of MSK. Methodology using descriptive statistical analysis was clearly presented. Results are also clearly presented, including tables and figures and also consistently supported by conclusion. Escalating burden of MSK was shown, while low back pain was seen as the most common issue. Significant geographic and gender disparities werea also present. Discussion deeply and scientifically provides great information and comparations between your study results and other studies. 

Author Response

  1. Interesting topic and, as you said - one of the first studies to describe the epidemiological aspect of musculoskeletal disorders in the USA. Introduction widely describes the entity and importance of MSK. Methodology using descriptive statistical analysis was clearly presented. Results are also clearly presented, including tables and figures and also consistently supported by conclusion. Escalating burden of MSK was shown, while low back pain was seen as the most common issue. Significant geographic and gender disparities werea also present. Discussion deeply and scientifically provides great information and comparations between your study results and other studies. 

Response: Thank you very much for your thoughtful and positive evaluation of our work. We appreciate your recognition of our study's significance in examining the epidemiological trends of musculoskeletal disorders, as well as your acknowledgment of the clarity in our methodology, results, and discussion sections. We are especially grateful for your comments on our focus on geographic and gender disparities, as well as our comparative analysis within the discussion, which we aimed to present in a way that meaningfully contextualizes our findings.

Reviewer 2 Report

Comments and Suggestions for Authors

Thank you for inviting me to review the manuscript entitled “Quantifying the strain: a global burden of disease (GBD) perspective on MSK disorders in the US over three decades 1990-2019”. 

The study has several strengths, including the use of a representative and big dataset, longitudinal evaluation, relevance of the topic, very explicative figures, the analysis of different risk factors and multidisciplinary domain of a healthcare condition.

The format of this manuscript does not adhere with the journal guidelines for the authors. A different template with page and line numbers should be utilized. 

Methods: 

I am relatively sure that the GBD assesses sex and not gender in their analysis. If sex was assessed (according to the type of question), then the authors should be consistent with the use of female / male and therefore sex throughout the study. For example, the tables read male / female, while the text read women / men. These terms are not interchangeable and the exact nomenclature should be used consistently. 

In the tables, there should be added a legend where the authors explain what a negative vs positive percentage mean, as well as explain what the two Percentage Change (%) signifies. 

It would be nice if the authors were also able to display the total % of MSK disorders across different US states in a heat map like the one of figure 3, maybe creating two panels of A and B where they show the change from 1990 to 2019.  

Discussion: 

When the authors provide the data of prevalence with numbers (e.g., 485,593 females vs 187,234 males, etc), it would be useful if they were able to provide prevalence ratio between the two sexes, instead of absolute numbers.

Attributing the higher prevalence of RA in females to the increased estrogen to androgen ratio is quite reductive, given that to begin with, females have a higher predilection for autoimmune disease due to the sex-dimorphism in immunological factors. The authors could also briefly add this further explanation. 

Author Response

Reviewer 2

Thank you for inviting me to review the manuscript entitled “Quantifying the strain: a global burden of disease (GBD) perspective on MSK disorders in the US over three decades 1990-2019”. 

  1. The study has several strengths, including the use of a representative and big dataset, longitudinal evaluation, relevance of the topic, very explicative figures, the analysis of different risk factors and multidisciplinary domain of a healthcare condition.

Response: Thank you for highlighting the strengths of our study, including our dataset, longitudinal approach, relevant topic, and the clarity of our figures. We appreciate your positive feedback on our analysis of risk factors and multidisciplinary approach, which we aimed to emphasize to provide a comprehensive perspective on this healthcare issue.

  1. The format of this manuscript does not adhere with the journal guidelines for the authors. A different template with page and line numbers should be utilized. 

Response: Thank you for this helpful note. We have edited our current submission to ensure all required elements are included per the journal's template. However, we plan to finalize the placement of text within the template pending further revisions and additional changes, especially as it relates to the current referencing format we have using a referencing software.

Methods: 

  1. I am relatively sure that the GBD assesses sex and not gender in their analysis. If sex was assessed (according to the type of question), then the authors should be consistent with the use of female / male and therefore sex throughout the study. For example, the tables read male / female, while the text read women / men. These terms are not interchangeable and the exact nomenclature should be used consistently. 

Response:  Thank you for your feedback regarding the use of sex and gender interchangeably. We carefully reviewed the manuscript and maintained consistency by using female/male in alignment with the GBD.

  1. In the tables, there should be added a legend where the authors explain what a negative vs positive percentage mean, as well as explain what the two Percentage Change (%) signifies. 

Response: Thank you for the helpful suggestion. We have added a legend underneath each table clarifying both the positive and negative percent change.

  1. It would be nice if the authors were also able to display the total % of MSK disorders across different US states in a heat map like the one of figure 3, maybe creating two panels of A and B where they show the change from 1990 to 2019.  

Response: Thank you for your helpful suggestion to include a heat map illustrating the total percentage of MSK disorders across different U.S. states over time. Given the existing number of figures in the manuscript, we decided to retain the current layout to maintain clarity and conciseness. However, we appreciate your idea and will certainly consider adding such visualizations in future work to enhance data presentation.

Discussion: 

  1. When the authors provide the data of prevalence with numbers (e.g., 485,593 females vs 187,234 males, etc), it would be useful if they were able to provide prevalence ratio between the two sexes, instead of absolute numbers.

Response: Thank you for your valuable suggestion, we have included the prevalence ratio for each MSK condition to complement the absolute numbers and provide a better understanding of the comparison.

  1. Attributing the higher prevalence of RA in females to the increased estrogen to androgen ratio is quite reductive, given that to begin with, females have a higher predilection for autoimmune disease due to the sex-dimorphism in immunological factors. The authors could also briefly add this further explanation. 

Response: Thank you for this insightful comment. We agree that attributing the higher prevalence of RA in females solely to the estrogen-to-androgen ratio is reductive. We have expanded the discussion to include sex-dimorphic immunological factors, such as heightened innate immune responsiveness and increased B-cell activity in females, which also contribute to the increased prevalence of autoimmune diseases. This addition provides a more comprehensive understanding of the factors influencing RA in females.

Reviewer 3 Report

Comments and Suggestions for Authors

Dear authors

I confess that when I started reading the manuscript “Quantifying the Strain: A Global Burden of Disease (GBD) Perspective on Musculoskeletal Disorders in the United States Over Three Decades, 1990 – 2019”, I was afraid that it would be something unoriginal and a reproduction of something that It is widely studied and disseminated. I also found it strange, given the topic, the heterogeneity of the research team, leading me to think that it was a fragment of a broader work, perhaps a part of any thesis of an academic work.

However, I was pleasantly surprised as I read on and realized that it is a very important approach to the topic, systematizing a set of information, about which we perhaps have a more partial view of the topic.

In this sense, I congratulate the authors on their approach. I believe that the manuscript is written simply and objectively, in any of the topics of the manuscript. Whether in the introduction, systematically going through the main components of the problem, or in terms of the methodology, presented in a very objective way, the same is true in the presentation of the results, structured, despite their density, they can differentiate and present in a simple way, focused on the main objectives of the study.

As for the discussion and conclusion of the results, I believe that they are equally well structured and objective, which, however, may leave out some questions that can be raised, based on the analysis of the results presented. As an example, what states that showed a decrease in the percentage of DALY age-standardized rate would have done differently, and why. However, I realize that it is not part of the objectives of this work.

Author Response

Reviewer 3

Dear authors

I confess that when I started reading the manuscript “Quantifying the Strain: A Global Burden of Disease (GBD) Perspective on Musculoskeletal Disorders in the United States Over Three Decades, 1990 – 2019”, I was afraid that it would be something unoriginal and a reproduction of something that It is widely studied and disseminated. I also found it strange, given the topic, the heterogeneity of the research team, leading me to think that it was a fragment of a broader work, perhaps a part of any thesis of an academic work.

However, I was pleasantly surprised as I read on and realized that it is a very important approach to the topic, systematizing a set of information, about which we perhaps have a more partial view of the topic.

In this sense, I congratulate the authors on their approach. I believe that the manuscript is written simply and objectively, in any of the topics of the manuscript. Whether in the introduction, systematically going through the main components of the problem, or in terms of the methodology, presented in a very objective way, the same is true in the presentation of the results, structured, despite their density, they can differentiate and present in a simple way, focused on the main objectives of the study.

As for the discussion and conclusion of the results, I believe that they are equally well structured and objective, which, however, may leave out some questions that can be raised, based on the analysis of the results presented. As an example, what states that showed a decrease in the percentage of DALY age-standardized rate would have done differently, and why. However, I realize that it is not part of the objectives of this work.

 Response: Thank you for your thoughtful and encouraging feedback. We are pleased that our approach to systematizing the data resonated with you and that you found our manuscript clear, focused, and well-structured across sections. We also appreciate your suggestion regarding exploring potential reasons behind variations in the DALY age-standardized rates across states. While this was beyond the scope of our current objectives, we recognize its value and may consider it in future analyses. Your insights are greatly appreciated and contribute positively to our work.